

# Simultaneous Detection of C₂H₆, CH₄ and δ¹³C-CH₄ Using Optical Feedback Cavity Enhanced Absorption Spectroscopy in the Mid-Infrared Region: Towards Application for Dissolved Gas Measurements

Loic Lechevallier[1,2], Roberto Grilli[1], Erik Kerstel[2], Daniele Romanini[2], Jérôme Chappellaz[1]

[1] CNRS, Univ. Grenoble Alpes, IRD, Grenoble INP, IGE, F-38000 Grenoble, France
[2] Univ. Grenoble Alpes, CNRS, LIPhy, F-38000 Grenoble, France

*Correspondence to*: Roberto Grilli (roberto.grilli@cnrs.fr)

**Abstract.** Simultaneous measurement of $C_2H_6$ and $CH_4$ concentrations, and of the $\delta^{13}C\text{-}CH_4$ isotope ratio is demonstrated using a cavity enhanced absorption spectroscopy technique in the mid-IR region. The spectrometer is compact and has been designed for field operation. It relies on optical-feedback assisted injection of 3.3-µm radiation from an Interband Cascade Laser (ICL) into a V-shaped high-finesse optical cavity. A minimum absorption coefficient of $2.8 \times 10^{-9}$ cm⁻¹ is obtained in a single scan (0.1 s) over 0.7 cm⁻¹. Precisions of 3 ppbv, 11 ppbv, and 0.08‰ for $C_2H_6$, $CH_4$, and $\delta^{13}C\text{-}CH_4$, respectively, are achieved after 400 s of integration time. Laboratory calibrations and tests of performance are reported here. They show the potential for the spectrometer to be embedded in a sensor probe for *in situ* measurements in ocean waters, which could have important applications for the understanding of the source and fate of hydrocarbons from the seabed and in the water column.

## 1 Introduction

Methane ($CH_4$) is the second most abundant anthropogenic greenhouse gas after carbon dioxide ($CO_2$), but with a 25 times higher global warming potential. Monitoring and identifying the different sources of $CH_4$ is therefore important, for the future climate projections. Among these sources, one can distinguish biogenic and abiogenic processes. Methane is mostly produced through biological processes, with the decomposition of organic matter under anoxic conditions. The abiogenic processes include biomass burning and thermal breakdown of organic molecules at high temperature in deep reservoirs. $CH_4$ is the main component of natural gas, but heavier hydrocarbons (HCs) can also be present in natural gas such as ethane ($C_2H_6$) and propane ($C_3H_8$), depending on the gas origin. In addition the carbon isotopic composition of methane ($\delta^{13}C\text{-}CH_4$ hereafter) differs between biogenic and abiogenic sources. While biogenic gas has high concentration of $CH_4$ with respect to heavier HCs and a more negative $\delta^{13}C\text{-}CH_4$ (typically from -60 to -90 ‰), thermogenic gas is characterized by a lower ratio of $CH_4/C_2H_6$ and a less negative $\delta^{13}C\text{-}CH_4$ signature (-50 to -40 ‰). The combination of these two measurements leads to a quite unambiguous identification of the origin of natural gas (Claypool and Kvenvolden, 1983). A significant part of earth's HC reservoirs lies in



marine environments, at variable depth below the seafloor. Questions arise about their origin and fate, and notably about their contribution through leakage into the ocean. Such leakage contributes to the carbon balance of the oceans, to acidification after oxidation in the water column, and possibly to the atmospheric concentration of these HCs if gas flaring from the seabed reaches the ocean surface. Therefore it is important to document the origin and fate of methane and ethane present below the

seabed and dissolved in the water column, for process understanding and for future climate and ocean acidity projections. Standard techniques for dissolved gas measurements usually rely on discrete sampling of the water column using Niskin bottles, followed by laboratory analysis. This method has the advantage of being simple and easy to setup, but it suffers from possible artefacts of the measurements due to outgassing of sample during the ascent (particularly for deep water samples). In addition, it offers a limited spatial and temporal resolution for probing the variability and mixing of water masses. Thanks to

significant efforts in miniaturization of analytical instruments, from mass spectrometers (MSs) to optical techniques, *in situ* measurements of dissolved gas composition are today feasible (Chua et al., 2016; Grilli et al., 2018; Wankel et al., 2013). Due to the complexity and size of isotopic ratio mass spectrometers (IRMSs), today the application of *in situ* MS is limited to the measurement of the abundance of dissolved gas species, while the isotopic analysis is most often still performed in the laboratory. On the other hand, modern optical spectrometers can offer similar performance for determining isotopic abundance

as MSs, while being more compact and easier to use for *in situ* operations. Cavity based techniques such as cavity ring-down spectroscopy (CRDS) and cavity enhanced absorption spectroscopy (CEAS) offer detection of a variety of species, multispecies detection, and the access to the isotopic signature of small compounds in gas mixtures (Kerstel, 2004).
The optical feedback cavity enhanced absorption spectroscopy (OFCEAS) (Morville et al., 2003, 2014) used in this work, is a well-known technique belonging to the  family of CEAS and widely employed in trace gas detection and related fields (Butler

et al., 2009; Grilli et al., 2014; Landsberg et al., 2014; Lechevallier et al., 2017; Richard et al., 2018; Romanini et al., 2006). It relies on the optical locking of the laser emission to a resonance mode of a high finesse optical cavity and it has already been demonstrated for distributed feedback (DFB) lasers, quantum cascade lasers (QCLs) (Gorrotxategi-Carbajo et al., 2013; Maisons et al., 2010), and interband cascade lasers (ICLs) (Manfred et al., 2015; Richard et al., 2016). In this work we report a novel application for the simultaneous measurement of $C_2H_6$, $CH_4$, and $\delta^{13}C$-$CH_4$, using an ICL radiation source in the mid-

IR region that could be applied to the *in situ* measurement of these compounds in seawater.

## 2 Method

The OFCEAS spectrometer (Figure 1) is composed of an ICL diode laser (from Nanoplus GmbH), an infrared polarizer (ColorPol MIR), two aluminum steering mirrors, and two photodiodes (PCI-2TE-4), all mounted on a stainless-steel optical cavity block. The V-shaped cavity (6-mm internal channel diameter) is composed of two 40-cm long arms, resulting in a free

spectral range (FSR) of the optical cavity of 187.4 MHz. The cavity mirrors (Lohnstar Optics) have a reflectivity of 99.9584%, which provides a ring-down time (lifetime of the photons inside the resonator) of 3.20 μs (~1km optical path length) in absence of intracavity absorption (i.e., if the cavity is filled with zero air). This design allows for feeding back to the laser only the



photons that are in resonance with the cavity, without any parasitic feedback, such as that from reflections off the entrance mirror. This modifies the laser dynamics, leading to spectral narrowing of the laser emission and forcing its oscillation to match the cavity resonance frequency (Morville et al., 2005). The optical feedback phase matching condition imposes that the laser is placed at a distance from the input mirror equal to an integer number of cavity-arm lengths (in this case equal to one).

Fine phase adjustment and continuous phase lock is obtained by a piezo-electric transducer (PZT) mounted on one of the two injection steering mirrors. The diode laser is stabilized in temperature and a current ramp is applied in order to produce a scan over a narrow frequency region (typically of a few tenths of nanometers, with a scanning rate of ~0.2 nm × mA$^{-1}$ for ICL lasers) and to temporarily lock the laser emission to successive $TEM_{00}$ longitudinal cavity transmission modes. The interest of the technique is to achieve high signal to noise ratio thanks to the narrowing of the laser emission and therefore to a more

efficient power coupling to the high-finesse cavity. It also provides an intrinsically linear frequency scale fixed by the cavity geometry ($FSR = c / (2\,N\,n\,L)$), where $c$ is the speed of light, $L$ the cavity arm length, $N$ the number of arms and $n$ the index of the media. The instrument is simultaneously recording the signal from the two photodiodes (before and after injection) in order to obtain transmission spectra that are subsequently converted to absorption units (absorption coefficients as a function of cavity transmission frequency or wavelength) using a measurement of the ring-down time performed at the end of each scan

(Morville et al., 2014). This ring-down event is produced by rapidly switching off the laser emission, and acquiring the exponential decay of the light leaking out of the cavity.

An OFCEAS spectrum acquisition covers more than 100 longitudinal cavity modes (0.7 cm$^{-1}$) and is repeated at a rate of 10 Hz. For a precise retrieval of the absorption profile, and therefore of the molecular density in the cell, the cavity is precisely temperature and pressure stabilized at respectively 308.15 K and 30 mbar. The cell is heated by two resistive heating bands

controlled via a PID regulation and a PT1000 sensor positioned at the center of the cell, glued on the stainless-steel block. A second PID loop is used for pressure regulation by acting on a solenoid proportional valve (FAS, Norgren Fluid Controls). The inner pressure in the optical cell is measured by a Wika, 0-250 mbar pressure gauge. The temperature and current of the laser and the temperature and pressure of the cell are controlled by a single electronic card developed by the company AP2E, which by means of a micro-processor, also ensures the fast acquisition of the two photodiode signals (340 kHz per channel). The

cavity with the optical assembly, reported in Figure 1, is suspended by two vibration-isolation dampers and placed in an insulated aluminium casing.

## 3 Spectral Region and Model Fit

In the infrared region, two fundamentals bands are available for detecting $CH_4$: the $\nu_3$ mode between 2800 and 3100 cm$^{-1}$ and the $\nu_4$ mode from 1200 to 1300 cm$^{-1}$. Several reasons make the $\nu_3$ transition more suitable for trace gas sensing, and notably:

(1) ICLs are today available at this wavelength and they require less effort to cool compared to room-temperature QCLs; and (2) photodetectors have better sensitivity in this spectral region. For this work, the selected laser is centred at 3.325 µm and operates between 8°C and 12°C, allowing it to cover the spectral region from 3005.4 to 3009.2 cm$^{-1}$. This region has been selected to best accommodate the absorption intensities of the species to be detected at a specific range of concentrations





(around 200 ppmv for $CH_4$ and 30 ppmv for $C_2H_6$) and for the possibility to access the 'best' isolated absorption lines. For applications in dissolved gas measurements, the selected wavelength region offers access to water absorption lines, required for the characterization of the dissolved gas extraction system (Grilli et al., 2018).

The selected absorption window reports a relatively large overlap of absorption features, requiring a rigorous optimization of

the fit parameters. Here, a multi-component fit routine is used where line parameters (intensity, position, Lorentzian and Gaussian widths) are fitted for each absorption transition (some of the parameters are linked together between lines of the same species to reduce the number of degrees of freedom of the spectral fit procedure). A total number of 46 absorption lines are included in the fit, with 17, 3, 23 and 3 lines for $^{12}CH_4$, $^{13}CH_4$, $C_2H_6$ and $H_2O$, respectively. A typical spectrum is reported in Figure 2 for concentrations of 72 ppmv of methane and 20 ppmv for ethane in dry air. The temperature and pressure of the cell

are 35°C and 30 mbar, respectively. The fit is optimized using an interlaced spectrum in order to increase the spectral resolution. The latter is obtained by slightly scanning the temperature of the cavity (0.02 °C of excursion), which causes a shift of the cavity mode positions with respect to the absorption lines, due to the mechanical elongation of the cell together with a change in the refractive index of the gas sample. For the spectrum of Figure 2 (experimental data: black and green lines), 200 consecutive spectra are interlaced to achieve a resolution of 935 kHz. In the top panel of the figure, the $^{12}CH_4$ and $^{13}CH_4$ (red

and blue spectrum, respectively) are the simulated spectra according to the HITRAN 2016 database (Gordon et al., 2017). This has been carried out for a clearer visualization of the line positions of the two isotopologues. A disagreement between experimental data and the HITRAN database was found for the two superposed lines of $^{12}CH_4$ at 3008.39 cm$^{-1}$, which are 16.5% weaker in amplitude in the simulated spectra with respect to the experimental data. The arrows indicate the best isolated absorption lines, which are used for calculating the isotopic ratio of methane (red and blue spectra) and the ethane concentration

(green spectrum). Optical interference fringes are visible on the baseline, with the major fringe corresponding to the optical path between the output cavity mirror and the photodiode. Improvement in the photodiode alignment and the use of an additional spherical mirror at the cavity output for focusing the light into the detector (as shown in Figure 1) significantly decreased the amplitude of the optical fringe.

The standard deviation of the fit residuals for the spectrum obtained with 72 ppmv of methane and 20 ppmv of ethane in dry

air is $5.8 \times 10^{-8}$ cm$^{-1}$ for a single acquisition, which corresponds to a precision of 140 ppbv, 1.3 ppbv, 37 ppbv and 1.5‰ for $^{12}CH_4$, $^{13}CH_4$, $C_2H_6$, and $\delta^{13}$C-CH$_4$, respectively. For the determination of the precision of $^{13}CH_4$, its natural abundance was taken into account. A contribution from small imperfections of the fit parameters is still present, since the standard deviation of the absorption baseline in absence of absorber (i.e. cavity filled with 30 mbar of zero air) is $2.8 \times 10^{-9}$ cm$^{-1}$ for a single acquisition. A more comparative way of describing the performance of the instrument consists of defining the figure of merit

normalized by the square-root of the bandwidth of the measurement (10 Hz) and the number of spectral elements (100) (Moyer et al., 2008). This leads to a figure of merit (Noise Equivalent Absorption Sensitivity, NEAS) of $8.8 \times 10^{-10}$ cm$^{-1}$ Hz$^{-1/2}$ per spectral element without account for fit imperfections and $1.8 \times 10^{-9}$ cm$^{-1}$ Hz$^{-1/2}$ per spectral element for a typical spectrum of 72 ppmv of $CH_4$ and 20 ppmv of $C_2H_6$ in dry air.



## 4 Results: The Performance of the Spectrometer

The stable carbon isotopic ratio ($\delta^{13}$C) is normally expressed as a comparative measurement with respect to an international reference, in this case Vienna Pee Dee Belemnite (VPDB; for the *Belemnitella Americana fossil carbonate)*. It is described by Eq. 1, in which $R$ is the abundance ratio of each isotopologue in the sample material, and $R_{VPDB}$ the corresponding abundance

ratio in the standard material. In the linear absorption regime, the number density of the molecule is proportional to the absorption and related to it via the absorption cross-section. Therefore, the retrieval of the isotope ratio with respect to a standard gas of known isotopic composition is relatively straightforward (Kerstel, 2004). The $\delta^{13}$C-CH$_4$ gas standards used for the experiments are from Isometric Instrument and certified on the VPDB scale with an uncertainty of ±0.2‰.

$$\delta^{13}C = \left(\frac{R}{R_{VPDB}} - 1\right) \qquad R = \frac{^{13}C}{^{12}C} \qquad (1)$$

Because of the complexity of the spectral fit and the discretization on the wavelength or frequency scale dictated by the cavity FSR, the position of the cavity modes with respect to one of the absorption lines affects the results of the fit. This is due to small imperfections of the fit model relative to the real spectrum which is sampled by the cavity modes. As these move and trace out the real spectrum profile, the model spectrum defects translate into varying errors or biases of the fit parameters which are optimized to match the model to the data at each time. We thus observed a dependency of the retrieved line surfaces

on the cavity mode positions with respect to the absorption features, which affects the isotope ratio determination (also observed and discussed in Favier, 2017). The drift in the cavity mode position is mainly due to mechanical instability and temperature fluctuations. Pressure may also play a role: even if stabilized at ±10 µbar the sensor may experience drifts with respect to the absolute pressure due to, for instance, fluctuations of temperature. In order to compensate for these drifts, we locked the cavity modes position by dynamically acting on the cavity temperature. Figure 3 reports long term measurements

for two situations: in red, where the temperature of the cavity is stabilized at 35 (±0.0015) °C and the cavity modes are free to move, and in black, where the cavity modes are locked by acting on the temperature setpoint (the shift reported corresponds to the MHz-size excursion of the cavity modes with respect to the absorption feature). Measurements were obtained by continuously flushing the cavity at a gas flow of 10 sccm (standard cubic centimeters per minute) with a sample containing 72 ppmv of CH$_4$ and 20 ppmv of C$_2$H$_6$ in dry air.

Without the locking of the cavity modes position we observed a drift of 20 MHz h$^{-1}$ of the cavity modes, while with the servo loop a stabilization of the mode positions better than 620 kHz was obtained. In the unlocked dataset, a fluctuation in the temperature stabilization occurred around 14h30 local time, and the corresponding section of data has been removed for the analysis. The spectral fit was conducted for both situations and an Allan-Werle (AW) statistical analysis (Werle, 2010) was performed on the $\delta^{13}$C-CH$_4$ signal (Figure 3). From the log-log plot, where the AW standard deviation ($\sigma_{AW-SD}$ $\delta^{13}$C) is plotted

against the averaging time (t), a tenfold improvement of the long-term stability was observed which translates into an extension of the optimum integration time of the system for the case of temperature locking of the cavity mode positions.



Afterwards, the system was further improved with a better optimization of the fitting parameters and of the optical setup resulting in a reduction of the optical interference fringes. A second AW statistical analysis was therefore conducted on the simultaneous detection of $\delta^{13}$C-CH$_4$, $^{12}$CH$_4$, and C$_2$H$_6$ with the same experimental conditions as for Figure 3. More than four hours of continuous measurements were used to produce the log-log plot (Figure 4). At short times, the measurement is mainly

dominated by white noise with the $\sigma_{AW-SD}$ decreasing proportional to $t^{-0.5}$. The optimum integration time corresponds to 400 s, reaching precisions of 0.08 ‰, 11 ppbv, and 3 ppbv for $\delta^{13}$C-CH$_4$, $^{12}$CH$_4$, and C$_2$H$_6$, respectively. For longer integration times, due to the rise of 1/f noise (drifts), the $\sigma_{AW-SD}$ starts to increase, leading to a degradation of the precision. In order to fix the accuracy of the instrument at its optimal precision below 0.1‰ for the $\delta^{13}$C-CH$_4$, injection of a standard gas with a known isotopic composition should thus occur every 400 s. This allows overcoming the instrumental drifts, and to maintain the

accuracy of the measurement at about the same level of precision obtained at the optimum integration time. If not, instrument drifts will dominate with a degradation of the accuracy as reported and explained below. A comparison of the two statistical analysis for the $\delta^{13}$C-CH$_4$ measurements (between Figure 3 and Figure 4) highlights how the improvement on the optical setup allowed to shift the entire AW curve to lower values (with a decreasing of the $\sigma_{AW-SD}$ by a factor of four), however we could not obtain the long integration time as reported in Figure 3. This is probably due to the fact that the arise of frequency dependent

noise is dominated by others instabilities, limiting the best precision of the $\delta^{13}$C-CH$_4$ to a value close to 0.1‰. Nevertheless, the benefit of the improvements is clearly visible, with the improved setup approaching the same precision than the previous one ten times faster.

In Figure 5, the dependency of the isotopic ratio with the concentration of methane is reported. This dependency is not caused by isotopic fractionation due to the sample handling, but more probably related to the discretization of the spectra and the

precise location of the measurement points with respect to residual optical fringes in the spectrum. The tendency is well reproducible, can be fitted with an exponential function and used for calibration propose. The same behaviour was also observed using the same technique but for H$_2$O and H$_2$S isotopic measurements (Favier, 2017; Landsberg, 2014), as well as with an instrument based on off-axis integrated cavity output spectroscopy (ICOS), as reported by Wankel et al., 2013. With a comparable amplitude variation of the absorption intensity, a similar effect (but less pronounced) was also observed with

respect to C$_2$H$_6$ concentrations. By monitoring the carbon isotopic ratio of methane with and without 22.5 ppmv of ethane in the gas mixture, a difference on the $\delta^{13}$C-CH$_4$ of 2.55 ‰ was observed. For this relatively small effect, a linear correction was applied.

The reproducibly of the system was estimated by conducting three measurements per day during four days at two methane concentrations (9 and 90 ppmv in dry air). Before each measurement the system was completely switched off. No calibration

using a standard gas was carried out before each measurement. Therefore, the resulting reproducibility corresponds to the intrinsic accuracy of the system affected by long term instabilities. Results are represented by the two histograms of Figure 5 showing the reproducibility for $\delta^{13}$C-CH$_4$, which is reported on a relative scale (i.e., with respect to an arbitrary reference value). At CH$_4$ concentrations of 9 and 90 ppmv the isotopic ratio is retrieved with an accuracy of ±11.8 ‰ and ±2.9 ‰,





respectively (both 1-$\sigma$). By injecting a standard gas before each measurement for calibration proposes, the accuracy at high concentrations should approach the expected precision of < 0.1 ‰ after 5 min of integration.

For the purpose of further validating the stability of the system, long term measurements were made while regularly switching between two samples with different isotopic composition. The two standard mixtures that were used had a composition of 79.4
ppmv of $CH_4$, with a carbon isotopic composition of -54.5 ‰, and of 56.9 ppmv with $\delta^{13}C$-$CH_4$ = -38.3 ‰ (with manufacturer certified uncertainties of ±0.2 ‰). The gas flow was fixed at 2 sccm and the standard gas samples were switched every five minutes during 3 hours using a manual three-way two-position valve. In Figure 6, the measurements of the carbon isotopic ratio are reported. Since the two samples had different $CH_4$ contents, a correction following the curve reported in Figure 5 was applied in order to account for the dependency of $\delta^{13}C$ on the $CH_4$ concentration. The -38.3 ‰ sample was employed as the
"known" standard gas, and the isotopic composition of the second sample was then retrieved. Drifts on the isotopic signature of only 0.6 ‰ were observed after 3 hours of continuous measurements (which is in agreement with the AW analysis). The response time of the instrument at 2 sccm gas flow is estimated to be 15 sec (single exponential), entirely attributed to the renewal of the gas sample in the measurement cell. The standard deviation of the raw measurement (at 10 Hz) is ± 1.9 ‰, while after averaging the results over 20 s (blue curve) the noise decreases to ±0.5 ‰.

**5 Comparison with existing instruments**

Depending on the application, different instrumental developments with a focus on a particular analytical aspect were carried out in the past. For instance, the concentration range will vary depending on the application: a high-sensitivity to measure in the low concentration range is required for environmental sensing (e.g. atmospheric measurements, paleoclimatology), while moderate sensitivity is required to measure in the high concentration range for dissolved gas measurements is seawater, leak
detection, and for oil and natural gas exploration. Some applications would require low sample volumes (such as ice-core and dissolved gas analyses). Finally, different constraints such as compactness, robustness, and power consumption, may play a role depending on the measurement strategy (laboratory, on site, or *in situ* measurements). For this propose, a non-exhaustive comparison with some of the existing techniques for measuring $\delta^{13}C$-$CH_4$ is reported in Table 1. The comparison highlights similar performances, albeit for different conditions, with precisions on $\delta^{13}C$-$CH_4$ below 1 ‰. Up to now, the best performance
in terms of precision and the amount of sample required remains IRMS technique, followed by the CRDS system developed by Picarro (Phillips et al., 2013; Schmitt et al., 2014). The QCLAS system of Eyer et al., 2016 based on a multi-pass cell, the ICOS from Los Gatos Research (Wankel et al., 2013), and the work reported here are more suitable for high methane concentrations. Our development aims at the measurement of dissolved gas in seawater, and therefore was optimized for relatively high methane concentrations of ~100 ppmv, achieving precision on the $\delta^{13}C$-$CH_4$ below 1 ‰ after 1 s. This
development sets itself apart by its compactness and portability as required for *in situ* measurements. Moreover, thanks to the low volume of the measurement cell, a small amount of sample (< 1 cm$^3$ of gas at STP) is required, making the spectrometer





suitable for coupling to a membrane-based extraction system for *in situ* application in the water column under marine conditions.

## Conclusion

In this work we presented a novel optical spectrometer, operating in the mid-IR region and based on the OFCEAS technique,
for simultaneous measurement of $C_2H_6$ and $^{12}CH_4$ concentrations, and the $\delta^{13}C$-$CH_4$. The spectrometer achieved a minimum absorption coefficient of $2.8 \times 10^{-9}$ cm$^{-1}$ over a single frequency scan (0.1 s) over 0.7 cm$^{-1}$ at low levels of $CH_4$ and $C_2H_6$ and $5.8 \times 10^{-8}$ cm$^{-1}$ for a more typical gas mixture (72 ppmv of $CH_4$ and 20 ppmv of $C_2H_6$). Precisions of 3 ppbv, 11 ppbv, and 0.08 ‰ for $C_2H_6$, $^{12}CH_4$, and $\delta^{13}C$-$CH_4$, respectively, were demonstrated for an optimum integration time of 400 s. A stability of $\pm$ 2.9 ‰ for long term measurements of $\delta^{13}C$-$CH_4$ was obtained without the need of regular injection of known isotopic
composition standard gas. The latter is needed if an accuracy of the $\delta^{13}C$-$CH_4$ measurement at the level of 0.1 ‰ is required. The dependency of the isotopic measurement on the composition of the gas mixture was characterized, with calibration curves for both $CH_4$ and $C_2H_6$ concentration dependencies. The instrument has the required characteristics of compactness, precision, and time resolution to be coupled with a dissolved gas extraction system and to be integrated in an *in situ* sensor for continuously monitoring the dissolved gas composition in deep-sea and oceanic environments. A dissolved gas extraction
system similar to the one described by Grilli *et al.*, 2018 will provide an adjustable dilution of the gas sample by adding a zero-air carrier gas at the dry side of the extraction unit, therefore increasing the dynamical range of the sensor.

## Acknowledgements

The research leading to these results has received funding from the European Community's Seventh Framework Programme
ERC-2015-PoC under grant agreement no. 713619 (ERC OCEAN-IDs). The work was made possible thanks to pioneering investigations conducted under the European Community's Seventh Framework Programme ERC-2011-AdG under grant agreement no. 291062 (ERC ICE&LASERS) and with support from SATT Linksium of Grenoble, France, and of the Service Partenariat & Valorisation (SPV) of the CNRS. The authors thank the AP2E company and particularly K. Jaulin, for exchanges regarding the spectrometer development.

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





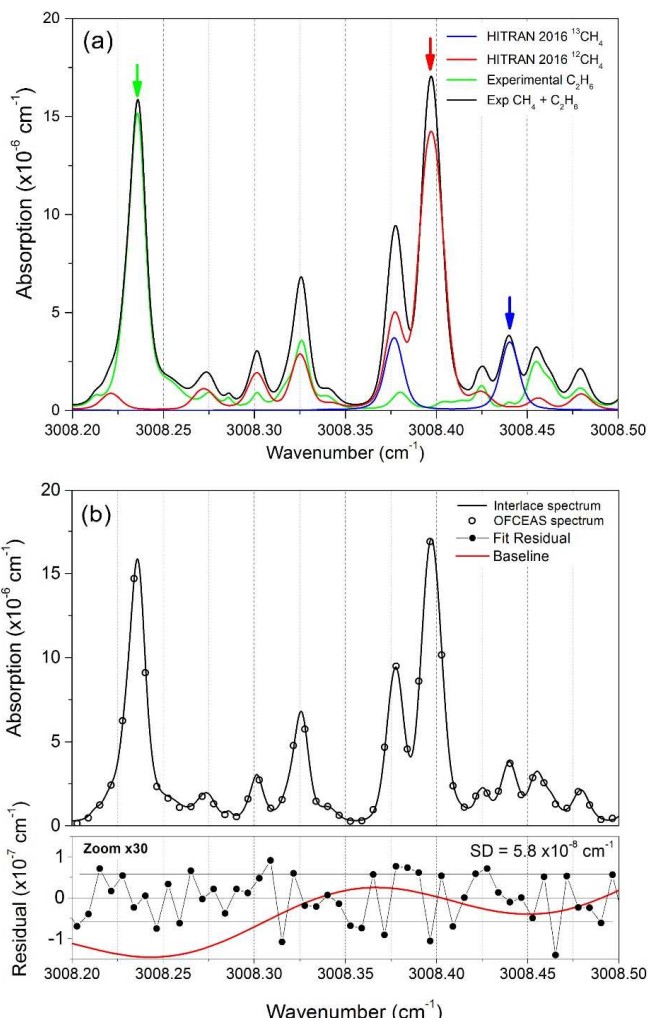

**Figure 2 :** **(a) A combination of experimental interlaced spectra and spectra simulated using the HITRAN 2016 database at 30 mbar and 35°C. HITRAN profiles show the position of $^{12}$CH4 (red) and $^{13}$CH4 (blue) at 72 ppmv of CH$_4$ in dry air. In green, the interlaced spectrum acquired with only 20 ppmv of C$_2$H$_6$ in dry air in the cavity. The full experimental spectrum containing absorptions from the three species is reported in black. There is a disagreement for the two superimposed lines at 3008.39 cm$^{-1}$, with a line intensity discrepancy between experimental data and the HITRAN 2016 database of ~16% on the amplitude of the absorption lines (this is outside the reported HITRAN 2016 uncertainties on the cross sections that are between 2% and 5%). (b) An example of an OFCEAS spectrum (black circles) and the corresponding interlaced methane-ethane absorption spectrum. At the bottom, the residuals of the spectral fit for one acquisition (0.1 s) and the baseline curve (red) are reported.**





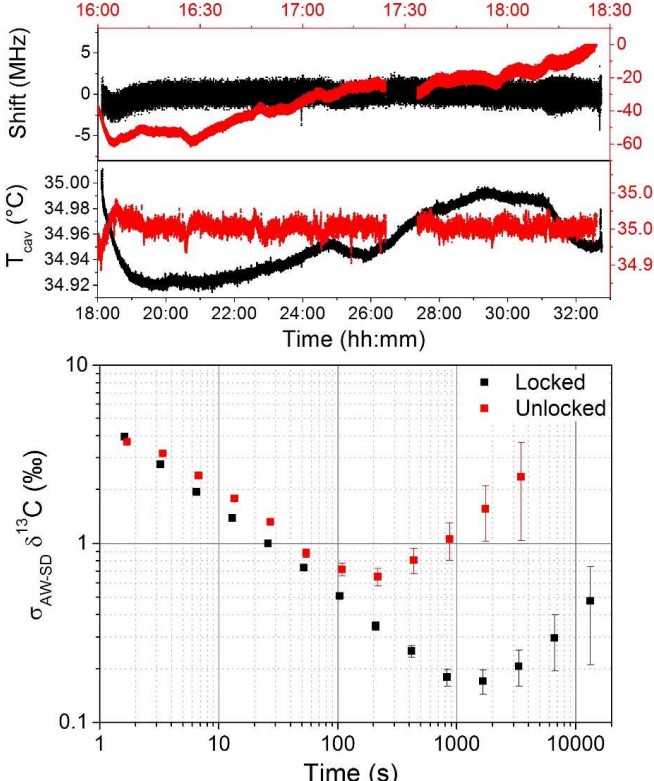

**Figure 3. A comparison of the performance on the measurement of the $\delta^{13}$C-CH$_4$ with (Black) and without (Red) the lock of the position of the cavity modes. Without the locking the temperature is stabilized at $35 \pm 0.0015$ °C and the mode position drift of about 20 MHz h$^{-1}$. By locking the mode positions by acting on the temperature setpoint, the mode positions are stabilized to 620 kHz. The advantage of the locking configuration is visible in the log-log plot, where the AW standard deviation of the $\delta^{13}$C-CH$_4$ signal is shifted to slightly lower values, with a longer averaging-time stability. Both measurements were performed by continuously flushing the cavity with a 10 sccm gas flow of a synthetic air mixture containing 72 and 20 ppmv of CH$_4$ and C$_2$H$_6$ in dry air, respectively.**




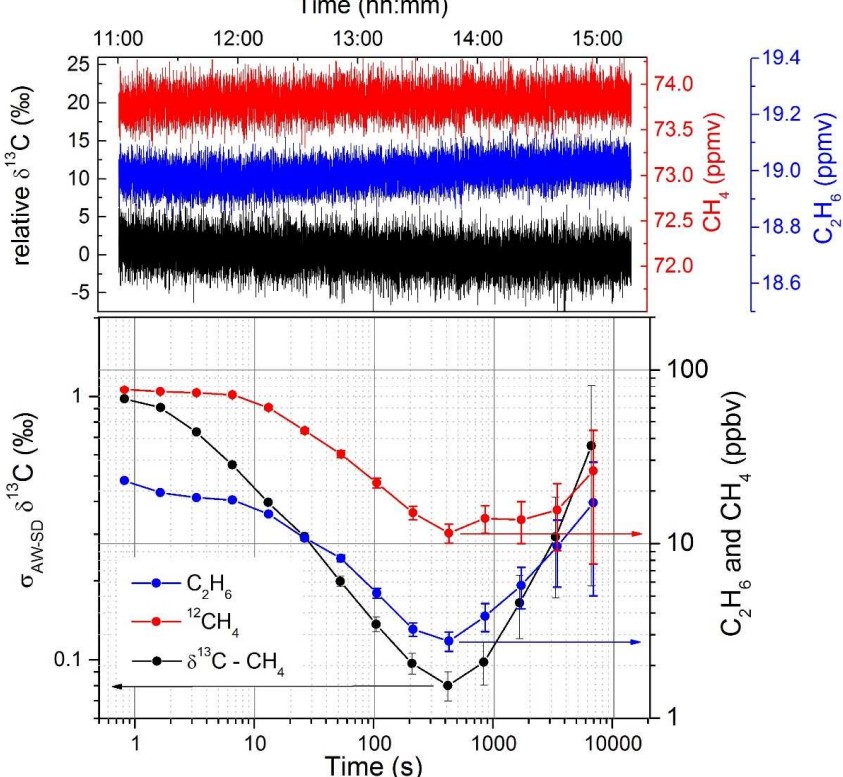

**Figure 4 : The Allan-Werle standard deviation for a four-hour time series of measurement with 73.7 ppmv of methane (black) and 19 ppmv of ethane (blue) in dry air. The best integration time of the system corresponds to ~8 min and the achieved precision is 0.08 ‰, 11, and 3 ppbv for $\delta^{13}C\text{-}CH_4$, $^{12}CH_4$, and $C_2H_6$, respectively.**



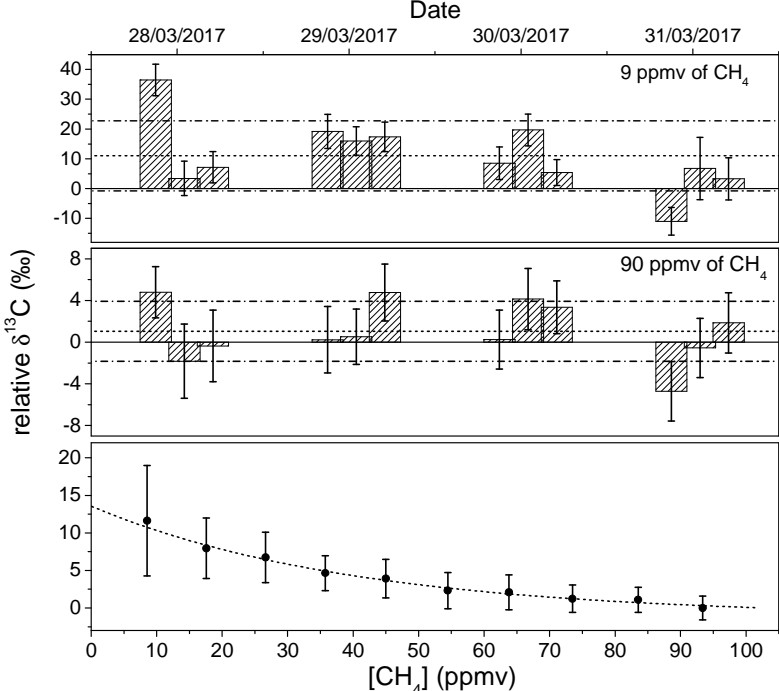

**Figure 5. Bottom panel: the dependency of the relative δ¹³C-CH₄ on the CH₄ concentration. Two upper panels: Reproducibility between discrete measurements made three times a day during one week for 9 and 90 ppmv of CH₄ in dry air. Accuracies of ±11.8 ‰ and ± 2.9‰ (represented by the dashed lines) were obtained for the two concentrations (1-σ). The instrument was not calibrated before each measurement using a gas standard. For high concentrations, such calibration would make the accuracy approaching the expected precision of 0.08 ‰ after 5 min of integration time.**



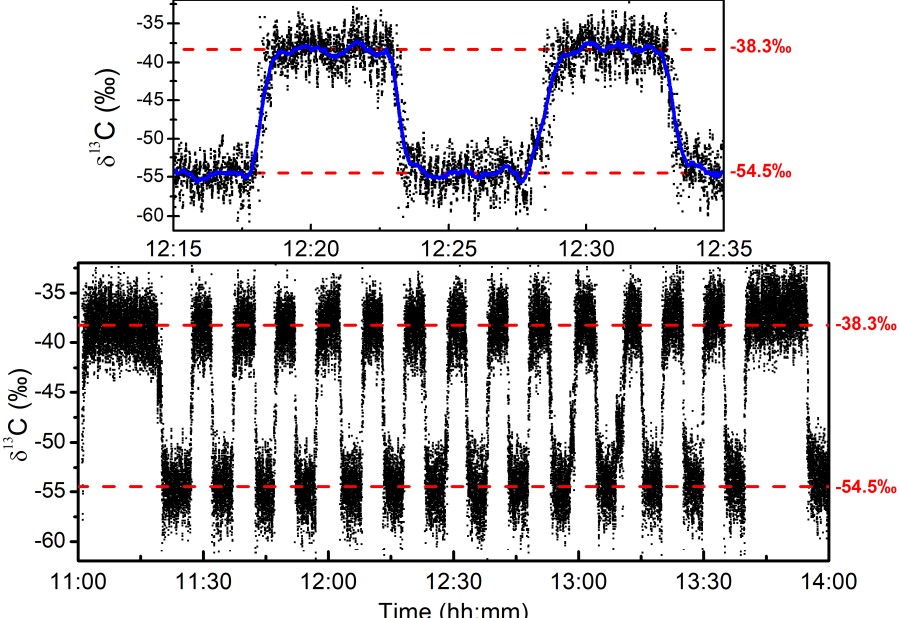

**Figure 6 : Long-term measurements obtained by repeatedly switching between two methane samples in dry air having different stable carbon isotopic composition. The two samples are diluted gas standards from Isometric Instrument with specified δ¹³C-CH₄ of -54.5±0.2‰ (79.4 ppmv) and -38.3±0.2‰ (56.9 ppmv).**





**Table 1. Non-exhaustive comparison between some of the existing techniques for measuring $\delta^{13}$C-CH$_4$**

| | Technique | Precision on the $\delta^{13}$C-CH$_4$ | Remarks |
|---|---|---|---|
| Eyer *et al.* 2016 | TREX-QCLAS (pre-concentration system coupled with a dual-QCL instrument with a multi-pass cell of 76 m) | 0.1 ‰ in 10 min (750 ppmv of CH$_4$) | Good performance at high concentration. It requires large gas volumes (0.5 L volume cell).Not adapted for application where low volumes of gas are available (e.g. ice-core, dissolved gases studies). |
| Wankel *et al.* 2013 | ICOS (off-axis integrated cavity output spectroscopy), Los Gatos Research, Inc. | 0.8 ‰ in 5 min (>1000 ppmv of CH$_4$) | Proved for applications with small amount of sample (i.e. deep water measurements). Designed for high concentrations. |
| Phillips *et al.* 2013 | CRDS (cavity ring-down spectroscopy), Picarro | 0.8 ‰ in 5 min (>1.8 ppmv of CH$_4$) | Designed for atmospheric measurements, it provides good reproducibility. Adapted to field measurements. |
| Schmitt *et al.* 2014 | IRMS (isotope ratio mass spectrometer), IsoPrime | 0.15 ‰ (atmospheric samples) | Best performances for low volume samples. Not adapted for field measurements. |
| This work | OFCEAS | 0.1 ‰ in 5 min (72 ppmv of CH$_4$) | Designed for *in situ* measurements and for low volume measurements. Degraded performances at low concentrations (atmospheric samples) |