# Peer review of "Simultaneous Detection of C2H6, CH4 and $\delta^{13}$ C-CH4 Using Optical Feedback Cavity Enhanced Absorption Spectroscopy in the Mid-Infrared Region: Towards Application for Dissolved Gas Measurements"

_Atmospheric Measurement Techniques, 2018_

## Referee Comment (RC1) · Anonymous Referee #1 · 28 Feb 2019

This paper deals with simultaneous detection of ethane, methane and carbon isotopic composition of methane by means of mid-IR optical feed-back cavity enhanced absorption spectroscopy. The performance of the spectrometer (as obtained in the laboratory environment) in terms of sensitivity and measurement precision makes it suitable for in-situ measurements in ocean waters. In this respect, the scientific motivations are strong and very well highlighted.

[Figure]

The work is performed well and the paper is clearly organized and written, with figures of excellent quality. Therefore, the manuscript deserves publication.

I recommend the authors to consider the comments reported hereafter.

1. The spectral region of interest shows a manifold of lines. The authors are obliged to consider 46 absorption lines in the fit of a single spectrum, with a relatively large number of free parameters. To mitigate this issue, physical constraints are applied to some of the parameters, thus reducing the degrees of freedom for the fitting procedure. Since this is one of the key points of the article, more details should be provided. In particular, a careful reader would like to know: the total number of free parameters; the number of experimental points of the interlaced spectrum; the adopted lineshape model.

2. This reviewer has the suspicious that the Voigt model has been used; if this is the case, I recommend the authors to give a look at the paper of Tennyson et al., Recommended isolated-line profile for representing high-resolution spectroscopic transitions (IUPAC Technical Report), Pure and Applied Chemistry, 2014. I am sure that the use of the HTP model would lead to reduced residuals. However, Figure 2 should provide also the residuals for the interlaced spectrum.

3. Temperature stabilization and control of the V-shaped cavity are requested in order to obtain a high-quality interlaced spectrum. In fact, the authors explain that an increase of the spectral resolution is achieved by slightly scanning the temperature of the cavity (0.02 °C of excursion), which causes a shift of the cavity mode positions with respect to the absorption lines. On page 3, they state that the cavity temperature is stabilized at 308.15 K, giving only two decimal digits. This means that the authors can control the temperature at the level of 10 mK, which is NOT sufficient for a refined control of the cavity modes. Nevertheless, reaching the mK level is surely not an easy task. Moreover, the requested equipment would limit the portability of the spectrometer. Such a key point should be discussed.

4. As for the dependence of the isotopic ratio on the methane concentration, this reviewer would suggest to consider the possibility of a side effect due to the choice of the wrong line shape model.

---

## Referee Comment (RC2) · Anonymous Referee #2 · 3 Mar 2019

This article presents an OF-CEAS based mid-IR interband cascade laser spectrometer for the simultaneous measurement of [CH4], [C2H6], and $\delta$13C-CH4. It is targeted to concentration ranges for CH4 and C2H6 as they are found in seawater. The manuscript is well written and structured, and the results are presented in a clear and concise fashion. I recommend publication in AMT after the authors have addressed a few minor points listed below.

[Figure]

page 2, line 27: in the text the ICL is said to be from Nanoplus, but in figure 1 you write NanoGiga.

page 3, line 6: there are a few different procedures to compute the signal used to steer the piezo-mounted mirror (to control the OF phase) (e.g. "asymmetry" of the modes etc.). Which one is used here?

page 4, line 8: what lineshape do you use for the spectral fit? Voigt? Could some of the issues mentioned later (i.e. dependence of $\delta$13C-CH4 on [CH4]) come from this choice?

page 4, line 14: after the 200 spectra are acquired, how are they interlaced? I.e. how do you know by how much the cavity modes of the n-th spectrum are offset from the (n-1)th spectrum?

page 4, line 31: is the exponent of your NEAS correct? should it not be -11 (per spectral point)?

page 5, line 14: what are line "surfaces"?

page 5, line 21: you say the positions of the cavity modes are "locked", but relative to what? To the time axis of each scan (i.e. the cavity modes should always occur at the same time relative to the start of the tuning ramp)?

page 5, line 27: I think you mean at 17h30 (not 14h30)

page 11, figure 1: solenoid (not solenoide)

page 12, figure 2: subscript the "4" in CH4 in the caption.

page 13, figure 3: you could use the same y-axis for the top panel (just a suggestion)

---

## Author Response (AR1)

We thank the reviewers for the very fruitful comments and remarks, which helped us to improve the manuscript.

All the remarks from the reviewers have been addressed below, and changes in the manuscript have been done accordingly (highlighted in red).

Reviewer(s)' Comments to Author:

**Referee #1:**

This paper deals with simultaneous detection of ethane, methane and carbon isotopic composition of methane by means of mid-IR optical feed-back cavity enhanced absorption spectroscopy. The performance of the spectrometer (as obtained in the laboratory environment) in terms of sensitivity and measurement precision makes it suitable for in-situ measurements in ocean waters. In this respect, the scientific motivations are strong and very well highlighted.

The work is performed well and the paper is clearly organized and written, with figures of excellent quality. Therefore, the manuscript deserves publication.

I recommend the authors to consider the comments reported hereafter.

(comment from Referee): The spectral region of interest shows a manifold of lines. The authors are obliged to consider 46 absorption lines in the fit of a single spectrum, with a relatively large number of free parameters. To mitigate this issue, physical constraints are applied to some of the parameters, thus reducing the degrees of freedom for the fitting procedure. Since this is one of the key points of the article, more details should be provided. In particular, a careful reader would like to know: the total number of free parameters; the number of experimental points of the interlaced spectrum; the adopted lineshape model.

(author's response) : The adapted lineshape model is the speed-dependent Rautian. This information has now be added in the manuscript (page 4) together with the references to the used model. More details about the number of points used for the fit and the degrees of freedom (9 in total) have been added in this section. "In order to reduce the degrees of freedom of the fitting routine and allowing a real-time fit for each acquired spectrum, some of the parameters have been pre-optimized and fixed or linked together. The remaining free parameters are: 1) the spectrum position: the positions of all lines are linked together and only one parameter (which corresponds to the mode-shift) is used to adjust the fit with respect to the acquired spectra; 2) the four intensities of $^{12}CH_4$, $^{13}CH_4$, $C_2H_6$ and $H_2O$ from which concentrations are retrieved, and all lines belonging to one of these molecules are fixed in relative intensity to pre-optimized values, with a single intensity scaling parameter for all of them; 3) the coefficients of the polynomial function for drawing the baseline of the fit (here corresponding to three parameters for a second order polynomial function); 4) one extra parameter can be added per optical fringes that the operator may want to fit. In this case we only have one optical fringe which has been identified between the output cavity mirror and the signal photodiode. The Gaussian and Lorentzian widths as well as the $\nu_{VC}$ parameter for Dicke narrowing are optimized line by line and then fixed to their optimum values. The total number of free parameters for the fit is 9 for a spectrum composed by 48 individual points."

(comment from Referee): This reviewer has the suspicious that the Voigt model has been used; if this is the case, I recommend the authors to give a look at the paper of Tennyson et al., Recommended isolated-line profile for representing high-resolution spectroscopic transitions (IUPAC Technical Report), Pure and Applied Chemistry, 2014. I am sure that the use of the HTP model would lead to reduced residuals. However, Figure 2 should provide also the residuals for the interlaced spectrum.

(author's response) : In the fit routine we used (the Postfit developed by D. Romanini) the HTP model was not developed. However, this should lead to similar results that the speed-dependent Rautian profile used in this work. The difficulty here is more related to the relatively high congestion of absorption lines, which make the fit optimization not trivial.
We report the residual of the interlaced spectrum below (in blue) and we adapted figure 2 in the manuscript accordingly. As one can see, the interlaced residual still have some structure related to possible artefacts of the spectra interlacing and some imperfection of the fit optimization. This interlaced spectra is only used for improving the fit parameters which are fixed. During normal operation of the instrument the discrete OFCEAS spectra (black dots) is used.

[Figure]

(comment from Referee): Temperature stabilization and control of the V-shaped cavity are requested in order to obtain a high-quality interlaced spectrum.  In fact, the authors explain that an increase of the spectral resolution is achieved by slightly scanning the temperature of the cavity (0.02°C of excursion), which causes a shift of the cavity mode positions with respect to the absorption lines. On page 3, they state that the cavity temperature is stabilized at 308.15 K, giving only two decimal digits. This means that the authors
can control the temperature at the level of 10 mK, which is NOT sufficient for a refined control of the cavity modes. Nevertheless, reaching the mK level is surely not an easy task. Moreover, the requested equipment would limit the portability of the spectrometer. Such a key point should be discussed.

(author's response) : This is indeed a key point that we now better explain in the manuscript. Reaching a temperature stability of the cavity better than 10 mK is challenging. By using a standard PID control and heating bands the temperature stability recorded by a PT1000 temperature probe are below 10 mK (the rms of the red curve in the top panel of Figure 3 is 1.5 mK). However, in this condition, the shift of the cavity mode position with respect to the absorption line positions (mode shift) reaches several tens of MHz. Since the composition of the gas in the cell (and therefore the refractive index) did not change during the measurement, this shift is most probably due to artefacts of the electronic, mainly in the temperature reading but possibly also in the pressure. In order to avoid the shift of the cavity modes, we decided to lock their position with respect to the absorption lines. The position of the absorption lines becomes therefore our reference for maintaining the cavity comb always in the same position with respect to the acquired spectra. This allows a stability of the comb below 1 MHz, while we let the temperature reading fluctuating (by few tens of mK as shown in Figure 3). At page 6 we now added: "The aim of the lock is not to better stabilize the cavity temperature, but to acting on this temperature regulation in order to maintain the cavity comb always at the same position with respect to the position of the absorption lines and compensate for electronic drifts occurring on the circuit for reading pressure and temperature of the cavity."

For further clearity we also added at page 4: "This interlacing is only used for fit optimization, while in normal operation the spectra are only composed by data points separated by the cavity FSR (187.4 MHz) with a total of 48 fitted spectral points. "

Regarding the temperature measurement, even if we can reach precision of 1.5 mK, on its absolute value we cannot have a better accuracy that 10 mK. That is why we state the temperature with only two digits. The temperature scans of 0.02°C are done in a relatively short time (<1 min), on which long term drifts mentioned above (due to artefacts by the electronic circuit) are negligible. Therefore this relative value of 0.02°C can be stated.  In the Labview routine employed for interlacing the spectra is only required to have a linear mode shift evolution oven more than one cavity FSR.

(comment from Referee):  As for the dependence of the isotopic ratio on the methane concentration, this reviewer would suggest to consider the possibility of a side effect due to the choice of the wrong line shape model.

(author's response) : As mentioned above, the speed-dependent Rautian fit was used in this work. We did not try the HTP model as suggested by the reviewer since the model is currently not implemented in the fir routine. Nevertheless, we think that we are limited by the high density of absorption lines in the used spectral region more than by the model used for the fit.

Reviewer(s)' Comments to Author:

**Referee #2:**

This article presents an OF-CEAS based mid-IR interband cascade laser spectrometer for the simultaneous measurement of [CH4], [C2H6], and δ13C-CH4. It is targeted to concentration ranges for CH4 and C2H6 as they are found in seawater. The manuscript is well written and structured, and the results are presented in a clear and concise fashion.  I recommend publication in AMT after the authors have addressed a few minor points listed below.

(comment from Referee): page 2, line 27: in the text the ICL is said to be from Nanoplus, but in figure 1 you write NanoGiga.

(author's response) : It is indeed an ICL form Nanoplus. The figure have been corrected.

(comment from Referee): page 3, line 6: there are a few different procedures to compute the signal used to steer the piezo-mounted mirror (to control the OF phase) (e.g.  "asymmetry" of the modes etc.). Which one is used here?

(author's response) : The phase locking is done by looking at the symmetry or the cavity mode. This has now be specified in the text "The phase error was retrieved by analyzing the symmetry of the cavity modes during the acquisition, as explained in (Morville et al., 2005). "

(comment from Referee): page 4, line 8: what lineshape do you use for the spectral fit? Voigt? Could some of the issues mentioned later (i.e. dependence of $\delta 13C-CH4$ on [CH4]) come from this choice?

(author's response) : Please see the answers to the Referee #1.

(comment from Referee): page 4, line 14: after the 200 spectra are acquired, how are they interlaced? I.e. how do you know by how much the cavity modes of the n-th spectrum are offset from the (n-1)th spectrum?

(author's response) : For the interlacing, we scan the cavity temperature by a bit more than one cavity FSR. For the interlacing to work well this scanning has to be as linear as possible in order to avoid artefacts in the spectrum. A dedicated labview routine is then identifying after how many consecutive spectra 1 FSR scan is achieved and it is homogenously distributing the spectra within this 1 FRS window. This has now been described in the manuscript "The interlacing is computed by a custom Labview routine that recognize after how many consecutive spectra a complete scan of the cavity FSR is complete. The spectra are then homogeneously distributed within the one FSR span to originate the interlaced spectrum."

(comment from Referee): page 4, line 31: is the exponent of your NEAS correct? should it not be -11 (per spectral point)?

(author's response) : The referee is correct. The values reported where not normalized by the root-square of the number of spectral elements. We further realized that the number of spectral elements used for the calculation should be 48 (as shown in figure 2). Therefore we now reported the correct values of $1.28 \times 10^{-10}$ and $2.6 \times 10^{-9}$ $cm^{-1}$ $Hz^{-1/2}$ per spectral element.

(comment from Referee): page 5, line 14: what are line "surfaces"?

(author's response) : It is the area underneath the absorption line. This is now specified in the manuscript (areas underneath the absorption line)

(comment from Referee): page 5, line 21: you say the positions of the cavity modes are "locked", but relative to what? To the time axis of each scan (i.e. the cavity modes should always occur at the same time relative to the start of the tuning ramp)?

(author's response) : The position of the cavity modes is locked with respect to the position of the absorption lines. This is now clearer in the manuscript.

(comment from Referee): page 5, line 27: I think you mean at 17h30 (not 14h30)

(author's response) : Yes. This has been corrected.

(comment from Referee): page 11, figure 1: solenoid (not solenoide)

(author's response) : Corrected (comment from Referee): page 12, figure 2: subscript the "4" in CH4 in the caption.

(author's response) : Corrected (comment from Referee): page 13, figure 3: you could use the same y-axis for the top panel (just a suggestion)

(author's response) : Figure 3 has been modified accordingly.

[revised manuscript text omitted]
 phase error was retrieved by analyzing the symmetry of the cavity modes during the acquisition, as explained in (Morville et al., 2005). The diode laser is stabilized in temperature and a current ramp is applied in order to produce a scan over a narrow frequency region (typically of a few tenths of nanometers, with a scanning rate of ~0.2 nm × mA$^{-1}$ for ICL lasers) and to temporarily lock the laser emission to successive TEM$_{00}$ longitudinal cavity transmission modes.

The interest of the technique is to achieve high signal to noise ratio thanks to the narrowing of the laser emission and therefore to a more efficient power coupling to the high-finesse cavity. It also provides an intrinsically linear frequency scale fixed by the cavity geometry ($FSR = c / (2\,N\,n\,L)$), where $c$ is the speed of light, $L$ the cavity arm length, $N$ the number of arms and $n$ the index of the media. The instrument is simultaneously recording the signal from the two photodiodes (before and after injection) in order to obtain transmission spectra that are subsequently converted to absorption units (absorption coefficients as a function of cavity transmission frequency or wavelength) using a measurement of the ring-down time performed at the end of each scan (Morville et al., 2014). This ring-down event is produced by rapidly switching off the laser emission, and acquiring the exponential decay of the light leaking out of the cavity.

An OFCEAS spectrum acquisition covers more than 100 longitudinal cavity modes (0.7 cm$^{-1}$) and is repeated at a rate of 10 Hz. For a precise retrieval of the absorption profile, and therefore of the molecular density in the cell, the cavity is precisely temperature and pressure stabilized at respectively ~308.15 K and 30 mbar. The cell is heated by two resistive heating bands controlled via a PID regulation and a PT1000 sensor positioned at the center of the cell, glued on the stainless-steel block. A second PID loop is used for pressure regulation by acting on a solenoid proportional valve (FAS, Norgren Fluid Controls). The inner pressure in the optical cell is measured by a Wika, 0-250 mbar pressure gauge. The temperature and current of the laser and the temperature and pressure of the cell are controlled by a single electronic card developed by the company AP2E, which by means of a micro-processor, also ensures the fast acquisition of the two photodiode signals (340 kHz per channel). The cavity with the optical assembly, reported in Figure 1, is suspended by two vibration-isolation dampers and placed in an insulated aluminium casing.

**3 Spectral Region and Model Fit**

In the infrared region, two fundamentals bands are available for detecting CH$_4$: the $\nu_3$ mode between 2800 and 3100 cm$^{-1}$ and the $\nu_4$ mode from 1200 to 1300 cm$^{-1}$. Several reasons make the $\nu_3$ transition more suitable for trace gas sensing, and notably: (1) ICLs are today available at this wavelength and they require less effort to cool compared to room-temperature QCLs; and (2) photodetectors have better sensitivity in this spectral region. For this work, the selected laser is centred at 3.325 µm and operates between 8°C and 12°C, allowing it to cover the spectral region from 3005.4 to 3009.2 cm$^{-1}$. This region has been selected to best accommodate the absorption intensities of the species to be detected at a specific range of concentrations (around 200 ppmv for $CH_4$ and 30 ppmv for $C_2H_6$) and for the possibility to access the 'best' isolated absorption lines. For applications in dissolved gas measurements, the selected wavelength region offers access to water absorption lines, required for the characterization of the dissolved gas extraction system (Grilli et al., 2018).

The selected absorption window reports a relatively large overlap of absorption features, requiring a rigorous optimization of the fit parameters. Here, a multi-component fit routine is used where line parameters (intensity, position, Lorentzian and Gaussian widths and the $\nu_{VC}$ parameter for collision-induced velocity change inducing a so called Dicke narrowing (Dicke, 1953)) are fitted for each absorption transition using a speed-dependent Rautian profile (Varghese and Hanson, 1984).

A total number of 46 absorption lines are included in the fit, with 17, 3, 23 and 3 lines for $^{12}CH_4$, $^{13}CH_4$, $C_2H_6$ and $H_2O$, respectively. In order to reduce the degrees of freedom of the fitting routine and allowing a real-time fit for each acquired spectrum, some of the parameters have been pre-optimized and fixed or linked together. The remaining free parameters are: 1) the spectrum position: the positions of all lines are linked together and only one parameter (which corresponds to the mode-shift) is used to adjust the fit with respect to the acquired spectra; 2) the four intensities of $^{12}CH_4$,

$^{13}CH_4$, $C_2H_6$ and $H_2O$ from which concentrations are retrieved, and all lines belonging to one of these molecules are fixed in relative intensity to pre-optimized values, with a single intensity scaling parameter for all of them; 3) the coefficients of the polynomial function for drawing the baseline of the fit (here corresponding to three parameters for a second order polynomial function); 4) one extra parameter can be added per optical fringes that the operator may want to fit. In this case we only have one optical fringe which has been identified between the output cavity mirror and the signal photodiode. The Gaussian and

Lorentzian widths as well as the $\nu_{VC}$ parameter for Dicke narrowing are optimized line by line and then fixed to their optimum values. The total number of free parameters for the fit is 9 for a spectrum composed by 48 individual points. A typical spectrum is reported in Figure 2 for concentrations of 72 ppmv of methane and 20 ppmv for ethane in dry air. The temperature and pressure of the cell are 35°C and 30 mbar, respectively. The fit is optimized using an interlaced spectrum in order to increase the spectral resolution. The latter is obtained by slightly scanning linearly the temperature of the cavity (0.02 °C of excursion), which causes a shift of the cavity mode positions with respect to the absorption lines, due to the mechanical elongation of the cell together with a change in the refractive index of the gas sample. This interlacing is only used for fit optimization, while in normal operation the spectra are only composed by data points separated by the cavity FSR (187.4 MHz) with a total of 48 fitted spectral points. The interlacing is computed by a custom Labview routine that recognize after how many consecutive spectra a complete scan of the cavity FSR is complete. The spectra are then homogeneously distributed within the one FSR

[revised manuscript text omitted]